# Learning an Efficient Convolution Neural Network for Pansharpening

**Yecai Guo \*, Fei Ye**  **and Hao Gong**

Jiangsu Key Laboratory of Meteorological Observation and Information Processing, Jiangsu Technology and Engineering Center of Meteorological Sensor Network, School of Electronic and Information Engineering, Nanjing University of Information Science and Technology, Nanjing 210044, China; feiye.chn@gmail.com (F.Y.); 20161242496@nuist.edu.cn (H.G.)

**\*** Correspondence: 001715@nuist.edu.cn; Tel.: +86-025-58731196

**Abstract:** Pansharpening is a domain-specific task of satellite imagery processing, which aims at fusing a multispectral image with a corresponding panchromatic one to enhance the spatial resolution of multispectral image. Most existing traditional methods fuse multispectral and panchromatic images in linear manners, which greatly restrict the fusion accuracy. In this paper, we propose a highly efficient inference network to cope with pansharpening, which breaks the linear limitation of traditional methods. In the network, we adopt a dilated multilevel block coupled with a skip connection to perform local and overall compensation. By using dilated multilevel block, the proposed model can make full use of the extracted features and enlarge the receptive field without introducing extra computational burden. Experiment results reveal that our network tends to induce competitive even superior pansharpening performance compared with deeper models. As our network is shallow and trained with several techniques to prevent overfitting, our model is robust to the inconsistencies across different satellites.

**Keywords:** pansharpening; convolutional neural network; nonlinear fusion model; dilated multilevel block; residual learning

## 1. Introduction

Motivated by the development of remote sensing technology, multiresolution imaging has been widely applied in civil and military fields. Due to the restrictions of sensors, bandwidth, and other factors, multiresolution images with a high resolution in both spectral and spatial domains are currently unavailable with a single sensor. Modern satellites are commonly equipped with multiple sensors, which measure panchromatic (PAN) images and multispectral (MS) images simultaneously. The PAN images are characterized by a high spatial resolution with the cost of lacking spectral band diversities, while MS images contain rich spectral information, but their spatial resolution is several times lower than that of PAN images. Pansharpening is a fundamental task that fuses PAN and MS images jointly to yield multiresolution images with the spatial resolution of PAN and the spectral information of the corresponding MS images.

Many research efforts have been devoted to pansharpening during the recent decades, and a variety of pansharpening methods have been developed [1,2]. Most of these methods can be divided into two categories, i.e., traditional algorithms and deep-learning-based methods. The traditional pansharpening methods can be further divided into three branches: (1) component substitution (CS) based methods, (2) multiresolution analysis (MRA) based methods, and (3) model-based optimization (MBO) methods. The CS-based methods assume that the spatial information of the up-sampled low-resolution MS (LRMS) lies in the structural component, which can be replaced

with the PAN image. Examples of CS-based methods are principal component analysis (PCA) [3], intensity-hue-saturation (IHS) [4], and Gram Schmidt (GS) [5], which tend to significantly improve the spatial information of LRMS at the expense of introducing spectral distortions. The guiding concept of MRA approach is that the missing information of LRMS can be inferred from the high-frequency content of corresponding PAN image. Hence, MRA-based methods, such as decimated wavelet transform (DWT) [6], Laplacian pyramid (LP) [7], and modulation transfer function (MTF) [8], extract spatial information with a corresponding linear decomposition model and inject the extracted component into LRMS. Pansharpening models guided by MRA are characterized by superior spectral consistency and higher spatial distortions. MBO [9–11] is an alternative pansharpening approach to the aforementioned classes, where an objective function is built based on the degradation process of MS and PAN. In this case, the fused image can be obtained via optimizing the loss function iteratively, which can be time-consuming.

All the above-mentioned methods fuse with linear models, and these methods cannot achieve an appropriate trade-off between spatial quality and spectral preservation, as well as computational efficiency [12]. To overcome the shortcomings of linear model, many advanced nonlinear pansharpening models have been proposed, and among them, the convolutional neural network (CNN) based methods, such as pansharpening by convolutional neural networks (PNN) [13], deep network architecture for pan-sharpening (PanNet) [14], and deep residual pansharpening neural network (DRPNN) [15], are some of the most promising approaches. Compared with the previously discussed algorithms, these CNN-based methods significantly improve the pansharpening performance. However, those pansharpening models are trained on specific datasets with deep network architecture, and when generalized to different datasets, they tend to be less robust.

In this paper, we adopt an end-to-end CNN model to address a pansharpening task, which breaks the linear limitation of traditional fusion algorithms. Different from most existing CNN-based methods, we have motivated our model as being more robust to the inconsistencies across different satellites. The contributions of this work are summarized as follows:

(1)  We propose a four-layer inference network optimized with deep learning techniques for pansharpening. Compared with most CNN models, our inference network is lighter and requires less power consumption. Experiments demonstrate that our model significantly decreases the computational burden and tends to achieve satisfactory performance.

(2)  To make full use of the features extracted by convolutional layers, we introduce a dilated multilevel structure, where the former features under different receptive fields are concatenated with a local concatenation layer. We also introduce an overall skip connection to further compensate the lost details. Experimental results reveal that with local and overall compensation, our multilevel network exhibits novel performance even with four layers.

(3)  As our network is shallow and trained with several domain-specific techniques to prevent overfitting, our model exhibits more robust fusion ability when generalized to new satellites. This is not a common feature of other deep CNN approaches, since most of them are trained on specific datasets with deep networks, which lead to severe overfitting problem.

## 2. Related Work

### 2.1. Linear Models in Pansharpening

The observed satellite imageries MS ($m$) and PAN ($p$) are assumed as degraded observations of the desired high-resolution MS (HRMS), and the degradation process can be modelled as:

$$\begin{cases} m = (x * k) \downarrow_4 + \varepsilon_{MS} \\ p = x * H + \varepsilon_{PAN} \end{cases} \tag{1}$$

where, $x * k$ represents the convolution between the desired HRMS ($x$) and a blurring kernel $k$, $\downarrow_4$ is a subsequent down-sampling operator with a scale of 4; $H$ is a spectral response matrix, which down-samples HRMS along the spectrum, $\varepsilon_{MS}$ and $\varepsilon_{PAN}$ are additive noise. Accordingly, the MBO method addresses pansharpening by forming an optimization function as:

$$\hat{x} = arg \min_{x} \left\{ \alpha \|y - k * x\|_2^2 + \beta \|G\left(p - \sum_{i=1}^{I} \omega_i x_i\right)\|_2^2 + \gamma \varphi(x) \right\} \tag{2}$$

in which, $\hat{x}$ and $y$ denote the pansharpened result and LRMS respectively; $I$ is the number of spectral bands of $x$, $x_i$ is the $i$-th band of $x$, and $\omega$ represents an $I$-dimensional probability weight vector that satisfies $\sum_{i=1}^{I} \omega_i = 1$ and indicates the linear nature of this model. $G$ is a spatial difference operator to focus on high-frequency content, and $\varphi$ denotes a prior term, which is used to regularize the solution space. The trade-off parameters $\alpha$, $\beta$, and $\gamma$ are used to balance the contribution of each term in the model.

For the CS-based pansharpening family, a low-resolution PAN image is formed by combining the available LRMS linearly. The generated low PAN image is transformed into another space, assuming the spatial structure is separated from spectral component. Subsequently, the extracted spatial information is replaced with the PAN image, and the fusion process is completed by transforming the data into the original space. A general formulation of CS fusion is given by:

$$\hat{x}_k = y_k + \psi_k \left(p - \sum_{i=1}^{I} \omega_i y_i\right) \tag{3}$$

in which, $\hat{x}_k$ and $y_k$ are the $k$-th band of $\hat{x}$ and $y$, and $\psi_k$ denotes the injection gain of the $k$-th band. For the MRA-based approach, the contribution of $p$ to $\hat{x}$ is achieved via linear decomposition, and the general form of MRA pansharpening method is defined as:

$$\hat{x}_k = y_k + \phi_k (p - p * h) \tag{4}$$

where $h$ is the corresponding decomposition operator, and $\phi_k$ denotes injection gain of the $k$-th band.

All the above-mentioned approaches extract structural information of the specific bandwidth from the PAN image with linear models and inject the extracted spatial details into the corresponding LRMS band. However, the spectral coverage of PAN and LRMS images are not fully overlapped and information extracted in a linear manner may lead to spectral distortions. Furthermore, the transformation from LRMS to HRMS is complex and highly nonlinear such that linear fusion models can rarely achieve satisfactory accuracy. In order to further improve the fusion performance, a nonlinear model is needed to fit the merging process. Therefore, deep-learning-based methods are taken into consideration.

*2.2. Convolution Neural Networks in Pansharpening*

Convolutional neural networks (CNNs) are representative deep learning models that have revolutionized both image processing and computer vision tasks. Given the similarity between single image super-resolution (SISR) and pansharpening, breakthroughs achieved in SISR have made profound influences on pansharpening. For example, PNN [13] and Remote Sensing Image Fusion with Convolutional Neural Network (SRCNN+GS) [16] are pioneering CNN-based methods for pansharpening, while the prototype of them is introduced from SRCNN [17], which is a noted SISR specific network. PNN makes some modifications upon SRCNN to make the three-layer network fit the domain-specific problem of pansharpening. Though impressive performance gains have been achieved, as the network of PNN is relatively simple, there is still plenty of room for improvement. SRCNN+GS also adopts SRCNN to perform pansharpening, and the difference is that SRCNN is employed as an enhancer to improve the spatial information of LRMS, and GS is applied for further improvement.

Inspired by the success of residual networks [18], the limitation of network capacity has been greatly alleviated, and researchers have begun exploring this avenue for pansharpening. For instance,

multiscale and multidepth convolutional neural network (MSDCNN) [12] is a novel residual learning based model that consists of a PNN and a deeper multiscale block. Owing to the deep network architecture, MSDCNN is able to fit more complicated nonlinear mapping and boost the fusion performance. However, the deep architecture of MSDCNN is intractable to be efficiently trained due to gradient vanishing and overfitting. This is extremely important for pansharpening, where the training data are often scarce as opposed to that of other computer vision applications.

## 3. Proposed Model

Given the restriction of limited training samples, we propose a moderate model for pansharpening. As our network is composed of only four layers, we adopt two concepts to further improve the efficiency of the network: the proposed dilated multilevel block and overall skip connection. The architecture of our proposed model is displayed in Figure 1, the dilated filter and dilated multilevel block are displayed in Figure 2a,b respectively.

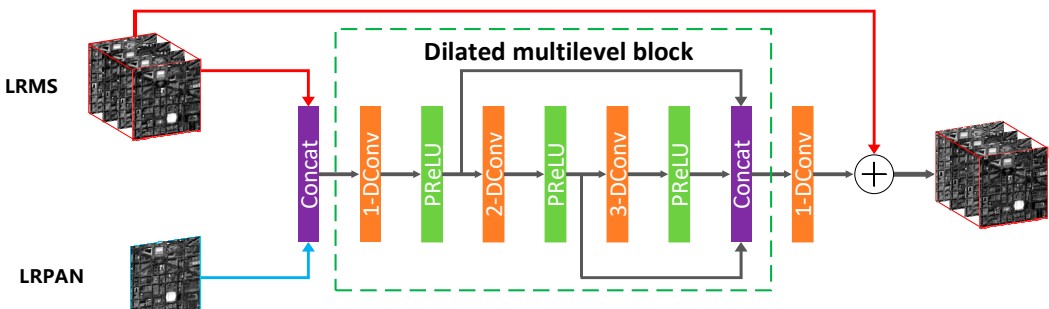

**Figure 1.** Architecture of the proposed dilated multilevel network.

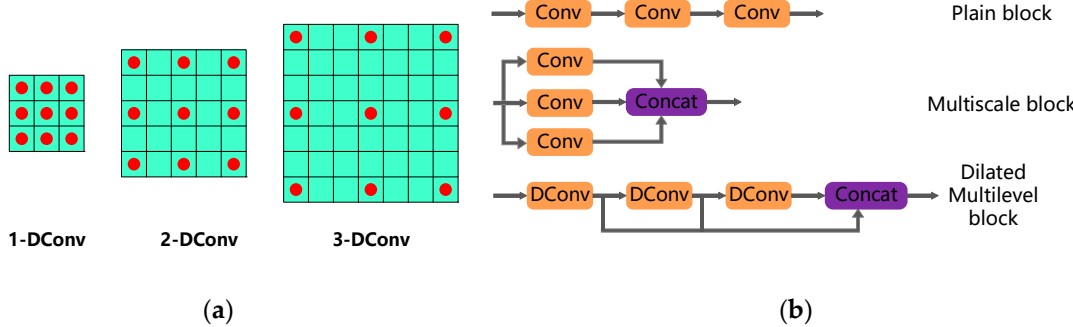

**Figure 2.** (**a**) Dilated filters with dilation factor s = 1, 2, 3. (**b**) Comparison of different block architecture.

### 3.1. Dilated Convolution

It has been commonly acknowledged that the context information facilitates the reconstruction of corrupted pixels in image processing tasks. To efficiently capture the context information, the receptive field of the CNN model is supposed to be enlarged during the training procedure. Specifically, the receptive field can be enlarged by stacking more convolutional layers or increasing the filter size; however, both of the two approaches significantly increase the computational burden and there is the risk of overfitting. As a trade-off between receptive field and network complexity, we adopt dilated convolution [19,20] as a substitute for traditional convolutional layer.

Dilated convolution is noted for its expansion capacity of the receptive field without introducing extra computational complexity. For the basic $3 \times 3$ convolution, a dilated filter with dilation factor s (s-DConv) can be interpreted as a sparse filter of size $(2s + 1) \times (2s + 1)$. The receptive field of the dilated filter is equivalent to $2s + 1$, while only 9 entries of fixed positions are non-zeros. Figure 2a

provides visualization of the dilated filter with dilation factors set as 1, 2, and 3. The complexity of our dilated multilevel block is calculated using:

$$O((I+1) \times 3 \times 3 \times C_{out} + C_{in} \times 3 \times 3 \times C_{out} + C_{in} \times 3 \times 3 \times C_{out}) \tag{5}$$

where $C_{in}$ and $C_{out}$ are the number of input and output channels of convolutional layers, which are set as 64 and the number of spectral bands ($I$) is 8. Without the dilated kernel, the computational complexity should be calculated using:

$$O((I+1) \times 3 \times 3 \times C_{out} + C_{in} \times 5 \times 5 \times C_{out} + C_{in} \times 7 \times 7 \times C_{out}) \tag{6}$$

It can be seen from Equations (5) and (6) that our method greatly reduces the cost of calculation by nearly 74%.

## 3.2. Dilated Multilevel Block

Since CNN models are formed by stacking multiple convolutional layers, as the network goes deeper, higher level features can be extracted, while lower structural details may be lost; Figure 3 provides further insight into this interpretation. By observing Figure 3b,c, we can find they match different features (vegetated areas and water basins) but tend to share similar structural information as that of LRMS, while higher level features in Figure 3d are more abstract compared with LRMS.

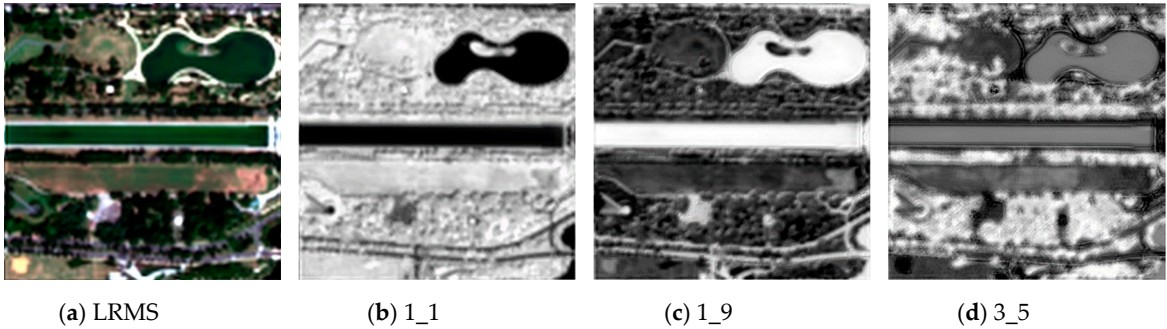

(**a**) LRMS        (**b**) 1_1        (**c**) 1_9        (**d**) 3_5

**Figure 3.** Input and intermediate results of a WorldView-2 sample. (**a**) Input of the CNN model. (**b**) The feature map obtained using filter #1 of the first convolutional layer. (**c**) The feature map obtained using filter #9 of the first convolutional layer. (**d**) The feature map obtained using filter #5 of the third convolutional layer.

To make full use of the extracted information, we propose the dilated multilevel block, which is introduced from multiscale block as displayed in Figure 2b. A multiscale block can learn representations of different scale under same receptive field, which can improve the abundance of extracted features, and has been applied in Reference [21]. Different from multiscale architecture, our proposed dilated multilevel block leverages both high- and low-level features sufficiently, which can make up the drawback of our lower network depth.

We compare the dilated multilevel block with the plain block and multiscale block to validate the superiority of the proposed one. All the experiments were conducted on same datasets and hyper-parameter settings. During the training procedure, all the networks are trained for $1.5 \times 10^5$ iterations and tested for 2000 iterations. Loss errors on the validation datasets are displayed in Figure 4a. As it shows, our dilated multilevel block outperformed plain block and multiscale block in improving pansharpening accuracy.

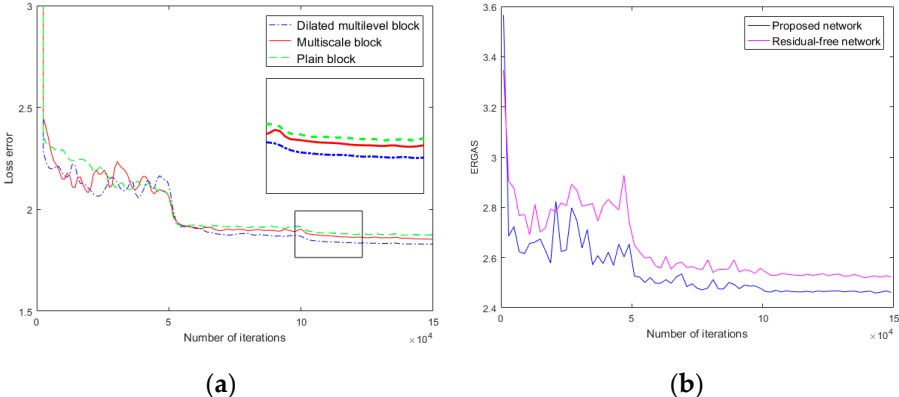

**Figure 4.** (**a**) Comparison of loss error on the validation dataset. (**b**) Performance-to-epoch curve of the proposed model and corresponding residual-free model.

### 3.3. Residual Learning

Convolutional layers are the core component of the CNN model, and with a deeper network, more complicated nonlinear mapping can be achieved, but the network tends to suffer from severe degradation problem. To overcome this problem, residual learning [18] is proposed and considered as one of the most effective solutions for training deep CNNs. The strategy of residual learning can be formulated as $H^m = \mathcal{R}(H^{m-1}) + H^{m-1}$, where $H^{m-1}$, $H^m$ are the input and output of the $m$-th residual block, respectively; and $\mathcal{R}$ denotes residual mapping. The residual mapping $\mathcal{R}$ learns the representation of $H^m - H^{m-1}$ rather than the desired target of the prediction $H^m$. With residual learning strategy, degradation caused by deep network can be significantly alleviated.

Residual representation can be formed by directly fitting a degraded observation to the corresponding residual component, like the ones employed in References [22,23]. Skip connection is another kind of technique to introduce residual representation, where it forms an input-to-output connection, which is employed in References [24,25]. In this paper, we used skip connection to introduce an overall residual learning strategy; with skip connection, lost details can be compensated for in the model. To further validate the efficiency of residual learning in the proposed model, we removed the skip connection from our CNN model, and denote the modified one as a residual-free network. We simulated the residual-free network with the same settings as those of the proposed one, and employ Erreur Relative Globale Adimensionnelle de Synthse (ERGAS) [26] for assessment; the performance-to-epoch curves are shown in Figure 4b. By observing Figure 4b, we can find the overall residual architecture achieves impressive performance gains.

## 4. Experiment

### 4.1. Experimental Settings

#### 4.1.1. Datasets

Our experiments were implemented on datasets from WorldView-2 and IKONOS respectively. Each of the datasets is sufficient to prevent overfitting, and some of them are available online (http://www.digitalglobe.com/resources/product-samples; http://glcf.umd.edu/data/ikonos/). Given the absence of HRMS at the original scale, the CNN model cannot be trained directly; as a conventional method, we followed Wald's protocol [27] for network training and experiment simulation. Specifically, we smoothed the MS and PAN with an MTF kernel [8,28] to match the sensor properties, and down-sample the smoothed component by a factor of 4. Subsequently, the degraded MS was up-sampled with bicubic interpolation to obtain LRMS; accordingly, the original MS image was regarded as HRMS. Figure 5 provides a pictorial workflow of training dataset generated based on Wald's protocol.

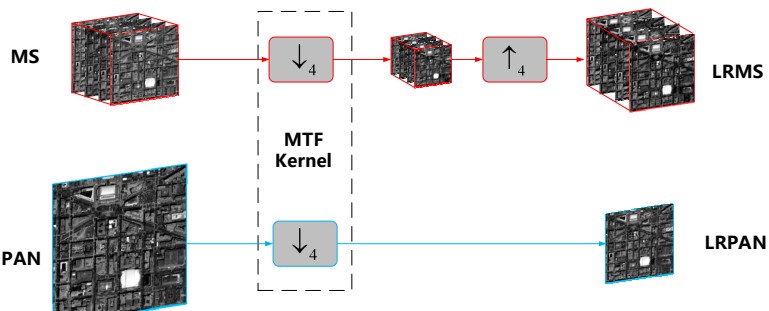

**Figure 5.** Generation of a training dataset through Wald protocol.

### 4.1.2. Loss Function

As mentioned in Section 3.3, an input-to-output skip connection was added to make up the lost details and perform residual learning; formally, the mapping function of our model is denoted as $\mathcal{R}(\boldsymbol{\omega}, \boldsymbol{b}; \{\boldsymbol{y}, \boldsymbol{p}\}, \boldsymbol{x}) = \boldsymbol{x} - \boldsymbol{y}$. Furthermore, the loss function is given as:

$$\widetilde{\mathcal{L}} = arg\ \min_{\boldsymbol{\omega}, \boldsymbol{b}} \mathcal{L}(\boldsymbol{\omega}, \boldsymbol{b}; \{\boldsymbol{y}, \boldsymbol{p}\}, \boldsymbol{x}) + \frac{\lambda}{2}\Omega(\boldsymbol{\omega}) \tag{7}$$

$$= arg\min_{\boldsymbol{\omega}, \boldsymbol{b}} \frac{1}{N} \sum_{l=1}^{N} \left( \frac{1}{2}\|\boldsymbol{x}^l - \mathcal{R}\left(\boldsymbol{\omega}, \boldsymbol{b}; \left\{\boldsymbol{y}^l, \boldsymbol{p}^l\right\}, \boldsymbol{x}^l\right) - \boldsymbol{y}^l\|_2^2 \right) + \frac{\lambda}{2}\|\boldsymbol{\omega}\|_2^2 \tag{8}$$

where $\mathcal{L}$ is a mean square error (MSE) term; $\boldsymbol{\omega}$ and $\boldsymbol{b}$ are weights and biases respectively; we let $\boldsymbol{\theta} = \{\boldsymbol{\omega}, \boldsymbol{b}\}$ represent all the trainable parameters in the model; $N$ is the batch size of the training datasets; and $\boldsymbol{x}^l$, $\boldsymbol{y}^l$, and $\boldsymbol{p}^l$ are the corresponding component of the $l$-th training sample in the batch. To further reduce the effect of overfitting, we employed a weight decay term ($\Omega$) to regularize the weights in the model, and $\lambda$ is the trade-off parameter.

The optimal allocation of $\boldsymbol{\theta}$ is updated iteratively with stochastic gradient descent (SGD) by calculating the gradients of $\widetilde{\mathcal{L}}$ to $\boldsymbol{\omega}$ and $\boldsymbol{b}$:

$$\begin{cases} \nabla_{\boldsymbol{\omega}}\widetilde{\mathcal{L}} = \nabla_{\boldsymbol{\omega}}\mathcal{L} + \lambda\boldsymbol{\omega} \\ \nabla_{\boldsymbol{b}}\widetilde{\mathcal{L}} = \nabla_{\boldsymbol{b}}\mathcal{L} \end{cases} \tag{9}$$

As the gradients obtained, we set a threshold as $\delta$, and clipped the gradients as [29]:

$$\left(\nabla_{\boldsymbol{\theta}}\widetilde{\mathcal{L}}\right)_{clipped} = \frac{\nabla_{\boldsymbol{\theta}}\widetilde{\mathcal{L}}\delta}{\max\left(\delta, \|\nabla_{\boldsymbol{\theta}}\widetilde{\mathcal{L}}\|_2^2\right)} \tag{10}$$

By clipping the gradients, the effect of gradient explosion can be removed.

To speed up the training procedure, we also adopted a classic momentum (CM) algorithm [30]. With the momentum and learning rate set as $\mu$ and $\varepsilon$, the updating of $\boldsymbol{\theta}$ is formed using:

$$\begin{aligned} \Delta\boldsymbol{\theta} &\leftarrow \mu \cdot \Delta\boldsymbol{\theta} - \varepsilon \cdot \left(\nabla_{\boldsymbol{\theta}}\widetilde{\mathcal{L}}\right)_{clipped} \\ \boldsymbol{\theta} &\leftarrow \boldsymbol{\theta} + \Delta\boldsymbol{\theta} \end{aligned} \tag{11}$$

### 4.1.3. Training Details

For each dataset, we extracted 83,200 patches for training and 16,000 patches for testing, where the size of training/validation patches were set to be $32 \times 32$. The learning phase of the CNN model was carried out on a graphics processing unit (GPU) (NVidia GTX1080Ti with CUDA 8.0) through the deep learning framework Caffe [31], and the test is performed with MATLAB R2016B configured with GPU. During the training phase, the loss function is optimized using SGD optimizer with the

batch size $N$ was set as 32. To apply CM and gradient clipping, $\lambda = 0.001$, $\delta = 0.01$, and $\mu = 0.9$ were used as default settings. Our CNN model was trained for $3 \times 10^5$ iterations and tested for per epoch (about 2000 iterations), with the initialized learning rate $\varepsilon$ set as $10^{-2}$. We updated the learning rate by dividing it by 10 at $10^5$ and $2 \times 10^5$ iterations. The training process of the proposed model cost roughly 4 h.

### 4.2. Experimental Results and Analysis

#### 4.2.1. Reduced Scale Experiment

In these experiments, the MS and PAN images were down-sampled by following Wald's protocol to yield the reduced scale pairs. In this case, we fused the degraded pairs and regarded the original MS as the ground truth. We merge our model on 100 images from WorldView-2 and IKONOS, and took that of WorldView-2 for detailed demonstration. Apart from the proposed CNN model, a number of state-of-the-art methods were also simulated for visual and quantitative evaluation. Specifically, we chose band-dependent spatial-detail (BDSD) [32], nonlinear intensity-hue-saturation (NIHS) [33], induction scaling technique based (Indusion) model [34], nonlinear multiresolution analysis (NMRA) [35], $\ell 1/2$ gradient based ($\ell 1/2$) model [36], PNN [13], and MSDCNN [12] for comparison. Among them, BDSD and NIHS belong to component substitution branch, Indusion and NMRA are MRA-based methods, and $\ell 1/2$ is guided by model-based optimization. For the deep-learning-based methods, PNN and MSDCNN are considered to be the main competitor of the proposed model.

Given the fact that the MS image of WorldView-2 contained eight spectral bands, we display the results composed of red, green and blue spectral bands (RGB spectral results) of one group in Figure 6 for visualization. To highlight the differences, we display the residual images in Figure 7 for a better visual inspection. For the numeric assessment, we employed the universal image quality index averaged over the bands (Q) [37], eight-band extension of Q (Q8) [38], spatial correlation coefficient (SCC) [39], Erreur Relative Globale Adimensionnelle de Synthse (ERGAS) [26], spectral angle mapper (SAM) [40], and feed-forward computation time for evaluation. The numeric indicators of simulated experiments are listed in Table 1.

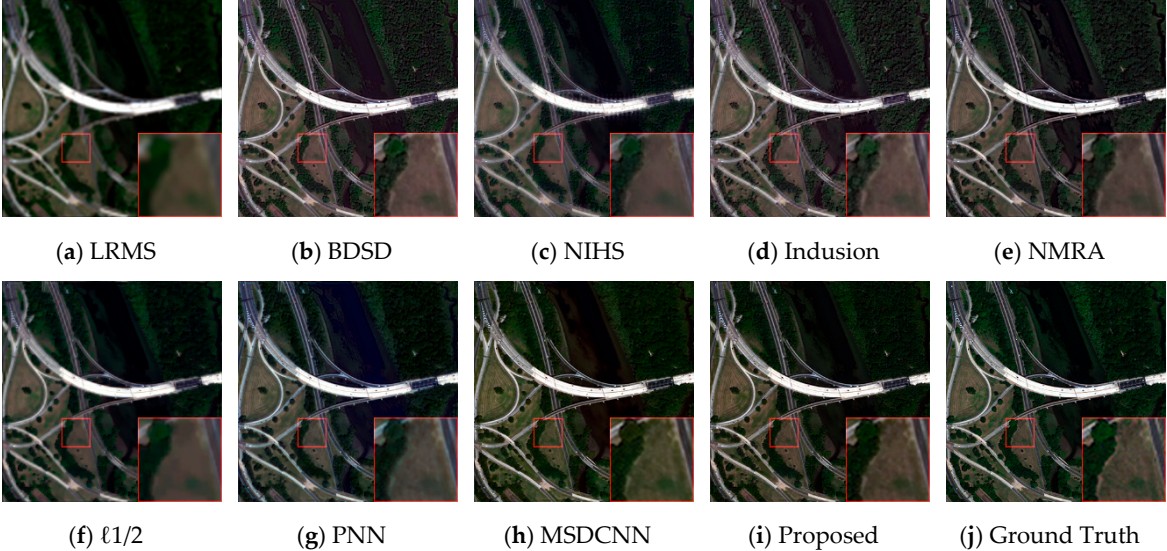

**Figure 6.** Results of the reduced scale experiment on an area extracted from a WorldView-2 image. (**a**) LRMS; (**b**) BDSD; (**c**) NIHS; (**d**) Indusion; (**e**) NMRA; (**f**) $\ell 1/2$; (**g**) PNN; (**h**) MSDCNN; (**i**) Proposed; (**j**) Ground Truth.

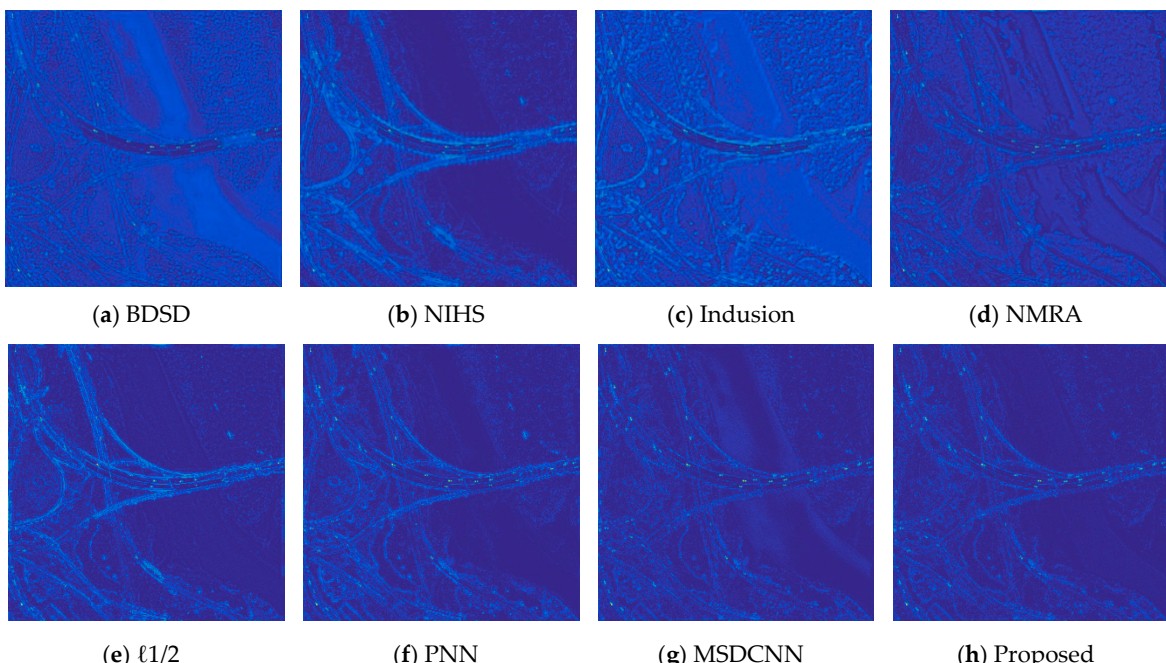

**Figure 7.** The residual images corresponding to Figure 6. (**a**) BDSD; (**b**) NIHS; (**c**) Indusion; (**d**) NMRA; (**e**) ℓ1/2; (**f**) PNN; (**g**) MSDCNN; (**h**) Proposed.

**Table 1.** Performance indicators of a WorldView-2 image at reduced scale.

|           | Q8     | Q      | SAM    | ERGAS  | SCC    | Time         |
|-----------|--------|--------|--------|--------|--------|--------------|
| Reference | 1      | 1      | 0      | 0      | 1      | 0            |
| BDSD      | 0.8224 | 0.8080 | 5.9732 | 4.0023 | 0.8145 | 0.25 s (CPU) |
| NIHS      | 0.7770 | 0.7642 | 5.1216 | 4.2357 | 0.7668 | 2.55 s (CPU) |
| Indusion  | 0.7948 | 0.7982 | 5.0844 | 3.8993 | 0.8016 | 0.17 s (CPU) |
| NMRA      | 0.8487 | 0.8413 | 4.5072 | 3.2280 | 0.8741 | 0.19 s (CPU) |
| ℓ1/2      | 0.8065 | 0.7880 | 4.7067 | 4.0404 | 0.7106 | 12.56 s (CPU)|
| PNN       | 0.8377 | 0.8459 | 5.0428 | 3.1775 | 0.9005 | 0.61 s (GPU) |
| MSDCNN    | 0.8741 | 0.8580 | 4.3776 | 2.7740 | 0.9149 | 0.14 s (GPU) |
| Proposed  | 0.8772 | 0.8758 | 3.7132 | 2.4658 | 0.9258 | 0.07 s (GPU) |

As we can observe from Figure 6, BDSB, NIHS, Indusion, and NMRA impressively improved the spatial details with the cost of introducing different levels of spectra distortions. In contrast, ℓ1/2 preserved precise spectral information, but the spatial components were rarely sharpened. Compared with the traditional pansharpening algorithms, the CNN-based methods tended to produce more satisfactory results. The proposed model effectively improved spatial information without introducing noticeable spectral distortions, while MSDCNN and PNN exhibit spectral distortions in specific regions (water basin). All these observations are also supported by the residual images displayed in Figure 7.

### 4.2.2. Original Scale Experiment

Since our CNN model is implemented at a reduced scale, we also fused the original LRMS and PAN for the sake of assessing the ability of transferring to original scale. Specifically, the raw MS image was up-sampled (LRMS) at the scale of PAN image, and we inputted the LRMS and corresponding PAN image into our model to yield full-resolution results. Same as the previous subsection, a typical example of WorldView-2 is displayed in Figure 8, and we also display the residual images in Figure 9. As the LRMS can be regarded as the low-pass component of HRMS, the optional residual in this section should be the high-frequency content of the desired HRMS, which means sharp edges without smooth regions.

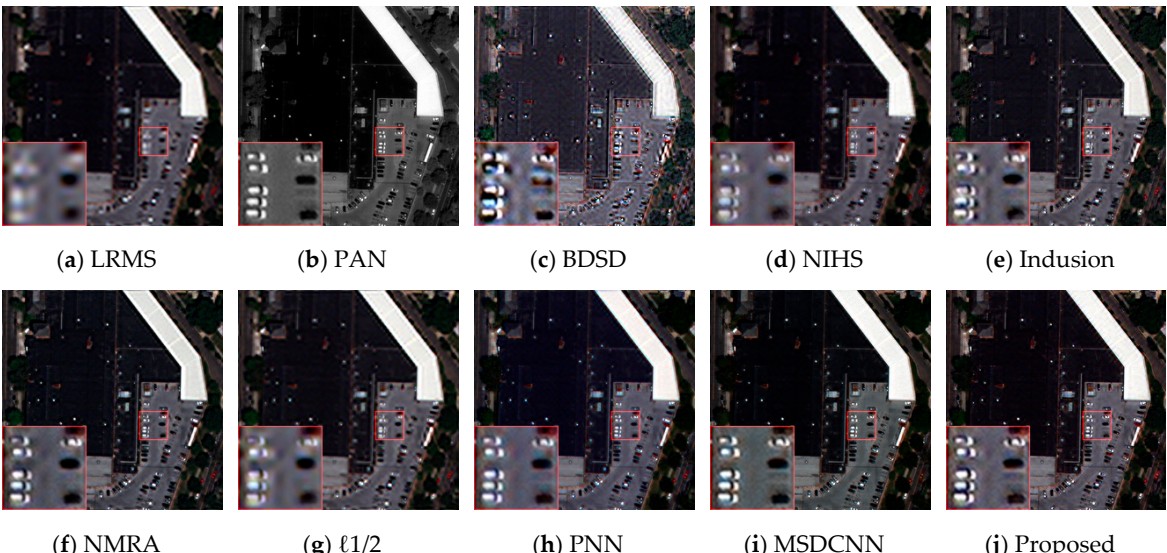

**Figure 8.** Results of the original scale experiment on an area extracted from a WorldView-2 image. (**a**) LRMS; (**b**) PAN; (**c**) BDSD; (**d**) NIHS; (**e**) Indusion; (**f**) NMRA; (**g**) $\ell 1/2$; (**h**) PNN; (**i**) MSDCNN; (**j**) Proposed.

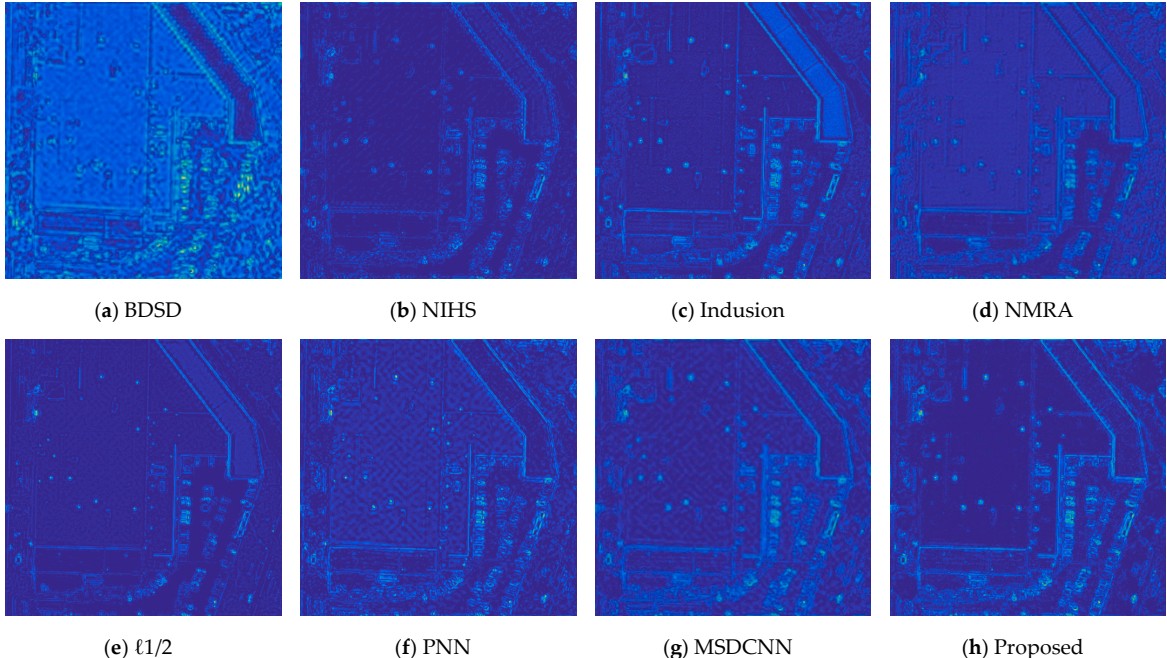

**Figure 9.** The residual images corresponding to Figure 8. (**a**) BDSD; (**b**) NIHS; (**c**) Indusion; (**d**) NMRA; (**e**) $\ell 1/2$; (**f**) PNN; (**g**) MSDCNN; (**h**) Proposed.

Given the absence of HRMS at original scale, three reference-free numeric metrics were adopted to quantify the qualities of fusion results, i.e., quality with no-reference index (QNR) [41], spectral component of QNR ($D_\lambda$) and spatial component of QNR ($D_S$). Apart from the three aforementioned non-reference metrics, we also followed Reference [42] by down-sampling the pansharpened results and compare the down-sampled results with the raw MS images. We tested the SAM and SCC indicators for spectral and spatial quality measurements, and the assessment results are summarized in Table 2.

**Table 2.** Performance indictors at original scale on WorldView-2 dataset.

|  | QNR | $D_\lambda$ | $D_S$ | SAM | SCC | Time |
|---|---|---|---|---|---|---|
| **Reference** | 1 | 0 | 0 | 0 | 1 | 0 |
| **BDSD** | 0.8609 | 0.0523 | 0.0916 | 3.9974 | 0.5944 | 0.24 s (CPU) |
| **NIHS** | 0.8566 | 0.0382 | 0.1094 | 2.1968 | 0.8098 | 2.99 s (CPU) |
| **Indusion** | 0.8359 | 0.0859 | 0.0855 | 1.9411 | 0.8112 | 0.16 s (CPU) |
| **NMRA** | 0.7453 | 0.1245 | 0.1486 | 1.8985 | 0.8196 | 0.50 s (CPU) |
| **$\ell1/2$** | 0.7813 | 0.0880 | 0.1423 | 1.8592 | 0.8083 | 12.89 s (CPU) |
| **PNN** | 0.8496 | 0.0434 | 0.1118 | 2.6279 | 0.8046 | 0.60 s (GPU) |
| **MSDCNN** | 0.8705 | 0.0397 | 0.0936 | 2.5754 | **0.8201** | 0.14 s (GPU) |
| **Proposed** | 0.9096 | 0.0197 | 0.0721 | 1.7561 | 0.8150 | 0.07 s (GPU) |

By comparing the images displayed in Figures 8 and 9, we can observe similar tendency as that of previous reduced scale experiments: NIHS and $\ell1/2$ preserve precise spectral information, while the spatial domains were rarely sharpened. Among the remaining results, BDSD introduces severe block artifacts, and Indusion and NMRA return images with competitive performance; however, when they come to residual analysis, we can find obvious spectral distortions. PNN and MSDCNN remain competitive in spatial details enhancement, while the proposed network performed better in preserving spectral details, which can be observed from the corresponding residual images.

### 4.2.3. Generalization

The design of our model was intended to be more robust when generalized to different satellites, as the proposed model was relative shallow and several improvements have been made to further boost the generalization. To empirically show this, we retained the model trained on WorldView-2 and IKONOS datasets to merge images from WorldView-3 QuickBird directly. We show the visual results in Figures 10 and 11.

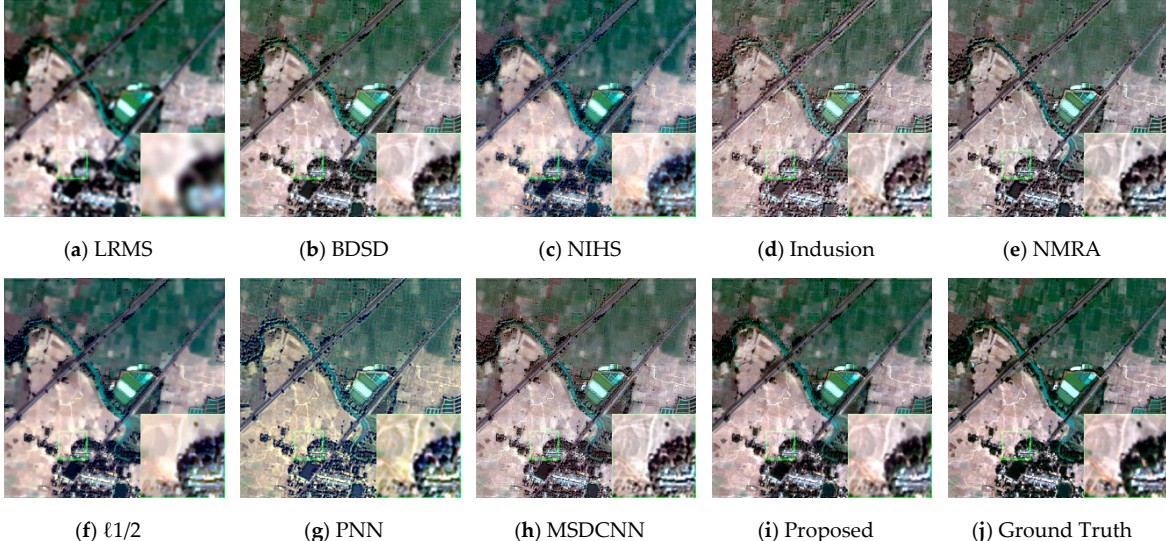

(**a**) LRMS    (**b**) BDSD    (**c**) NIHS    (**d**) Indusion    (**e**) NMRA

(**f**) $\ell1/2$    (**g**) PNN    (**h**) MSDCNN    (**i**) Proposed    (**j**) Ground Truth

**Figure 10.** Results of the reduced scale experiment on an area extracted from a QuickBird image. (**a**) LRMS; (**b**) BDSD; (**c**) NIHS; (**d**) Indusion; (**e**) NMRA; (**f**) $\ell1/2$; (**g**) PNN; (**h**) MSDCNN; (**i**) Proposed; (**j**) Ground Truth.

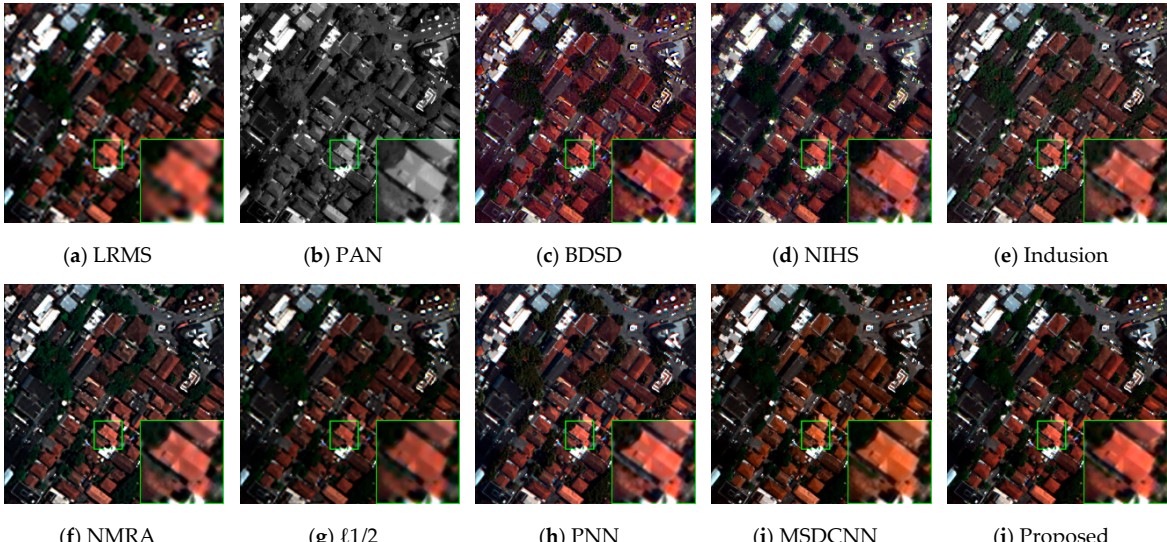

**Figure 11.** Results of the original scale experiment on an area extracted from a WorldView-3 image.
(**a**) LRMS; (**b**) PAN; (**c**) BDSD; (**d**) NIHS; (**e**) Indusion; (**f**) NMRA; (**g**) ℓ1/2; (**h**) PNN; (**i**) MSDCNN;
(**j**) Proposed.

As we can inspect from the fusion result, the proposed CNN model displayed stable performance with sharp edges and inconspicuous spectral distortions, while the other CNN models could not generalize well. Since PNN and MSDCNN neglected the effect of overfitting, MSDCNN in particular adopts a relative deep network with limited training samples, which makes the model less robust. For the remaining traditional algorithms, accordant conclusions as those of aforementioned experiments can be drawn from Figures 10 and 11: NIHS, Indusion, and NMRA introduce different levels of blurring artifacts, whereas ℓ1/2 and BDSD suffer from serious spectral distortions.

## 5. Conclusions

In this paper, we proposed an efficient model motivated by three goals of pansharpening: spectral preservation, spatial enhancement, and model robustness. For the spectral and spatial domains, we employed an end-to-end CNN model that breaks the limitation of linear pansharpening algorithms. Experimental results demonstrate that our model tended to return well-balanced performance in spatial and spectral preservation. For the improvement of robustness, our CNN model is shallow while efficient, which can be less prone to overfitting. By adopting dilated convolution, our model achieved a larger receptive field, and greatly made up the shortcoming of network depth. Among the model, we also employed a multilevel structure to make full use of the features extracted under different receptive fields. Compared with state-of-the-art algorithms, the proposed model makes a better trade-off between spectral and spatial quality as well as generalization across different satellites.

Our model was motivated by the aim of being robust when generalized to new satellites, which was designed under the guiding concepts of simplicity and efficiency. As the generalization was improved, the shallow network architecture also restricted the fusion accuracy. In our future work, we will stick to the fusion of one specific satellite dataset; in that case, we can remove the effect of generalization and focus on optimizing the architecture of a deeper network to further boost the fusion performance.

**Supplementary Materials:** The following are available online at http://www.mdpi.com/1999-4893/12/1/16/s1.

**Author Contributions:** Conceptualization, F.Y. and H.G.; methodology, F.Y.; software, H.G.; validation, F.Y., and H.G.; formal analysis, H.G.; investigation, F.Y.; resources, F.Y.; writing—original draft preparation, F.Y.; writing—review and editing, F.Y.; visualization, H.G.; supervision, Y.G.; project administration, Y.G.; funding acquisition, Y.G.

**Funding:** This research was funded by the National Natural Science Foundation of China under Grant 61673222.

**Conflicts of Interest:** The authors declare no conflict of interest.

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
