# Peer review of "Learning an Efficient Convolution Neural Network for Pansharpening"

_algorithms, doi:10.3390/a12010016_

Round 1
Reviewer 1 Report
The authors propose a dilated multilevel block based CNN to perform Pansharpening. The proposed model is claimed to achieve promising results than the state of the art. This reviewer has a few queries:
In the Introduction, the authors shall provide citations for Pansharpening. The authors claim that linear methods do not achieve a trade-off between spatial quality, computational efficiency, and spectral preservation. The statement has to be supported with substantial evidence.
How do the authors settle for the architecture of the dilated multilevel block? What is the effect of adding or removing layers? In other words, how did the authors optimize the architecture and hyperparameters of the proposed model?
What is the performance of the multiscale model with additional layers? Are these results purely empirical? Did they perform any optimization to learn the optimal network architecture and hyperparameters that gives the best performing model?
How did the authors perform the train/test split? Did the patches from the same image make it to the validation set as well? The authors shall perform k-fold cross-validation to generalize model performance.
Words are repeatedly abbreviated in the manuscript and need to be modified.
What is the size of the original image? Why do the authors settle for a patch size of 32×32? Did the authors perform any image preprocessing/data augmentation during model training?
Figure 4a lacks clarity. The authors shall use different colors to highlight the performance of plain and dilated blocks.
What is the effect of downsampling the MS and PAN images on the model accuracy?
Did the authors measure the log spectral distance to substantiate the claim that the proposed method introduces less spectral distortion?
Line 317: rest: test? Serve: severe?
What is the performance of the proposed method in comparison to the state of the art? The author shall test the performance with the benchmark datasets and provide a measure of the performance to compare with the state of the art.
Author Response
Please refer to the "Report Notes" for detailed review response.

Reviewer 2 Report
This paper is devoted to a very important problem in the area of processing of satellite imagies namely pansharpening. The aim of this task to enhance the spatial resolution of the multispectral image by fusing it with a corresponding panchromatic one. An efficient non-linear model is proposed that employs an end-to-end CNN model that breaks the limitation of linear algorithms. The model adopts dilated deconvolution operation and multilevel structure to achieve larger receptive field and to make full use of the extracted features. Compared with state-of-the-art algorithms, the proposed model makes a better trade-off between spectral and spatial quality as well as generalization across different satellites.
The paper definitely deserves publication but presentation is to be imroved. Please fix the following issues:
Even though all abbreviations are carefully supplied with references it would be beneficial to give full names when first mentioned, e.g. PNN [12], MSDCNN [13] and DRPNN [14] (line 60)
Fig 1 - full names of blocks are to be given in caption
Fig 3 should be Fig 2 as it is referenced after Fig 1
Summation with index i in formaula 5 line 159 is not clear as there is no index i inside
line 218: "reference-free" but figure: "residual-free"
m is not defined in line 200
line 214: "impressive performance gains" are to be calculated and given in numbers
line 222: WorldView-2 but WorldVorld-2 in line 168
line 239 defines L with tilde as MSE but there is also L without tilde in 7 that is MSE.
"max" in formula 10 line 248 should not be italized.
Author Response
Response to Reviewer 2 Comments
Dear reviewer,
Please find our revision to Algorithm, “learning an efficient convolution neural network for pansharpening”, with manuscript number: algorithms-412537.
We have made revision to the paper taking into account the reviewer’s comments. Please find our point-by-point response to the comments from the next page.
We thank the reviewer for helpful comments about our approach. We have tried to address everything satisfactorily in this revision.
Thanks,
Fei Ye
Corresponding Email: feiye.chn@gmail.com
This paper is devoted to a very important problem in the area of processing of satellite imagies namely pansharpening. The aim of this task to enhance the spatial resolution of the multispectral image by fusing it with a corresponding panchromatic one. An efficient non-linear model is proposed that employs an end-to-end CNN model that breaks the limitation of linear algorithms. The model adopts dilated deconvolution operation and multilevel structure to achieve larger receptive field and to make full use of the extracted features. Compared with state-of-the-art algorithms, the proposed model makes a better trade-off between spectral and spatial quality as well as generalization across different satellites. The paper definitely deserves publication but presentation is to be imroved. Please fix the following issues:
Point 1: Even though all abbreviations are carefully supplied with references it would be beneficial to give full names when first mentioned, e.g. PNN [12], MSDCNN [13] and DRPNN [14] (line 60).
Response 1: All abbreviations have been given full name when first mentioned in line 60 and line 274 of the revised manuscript.
Point 2: Fig 1 - full names of blocks are to be given in caption.
Response 2: The caption of Figure 1 has been fixed in line 145 of the revised manuscript.
Point 3: Fig 3 should be Fig 2 as it is referenced after Fig 1.
Response 3: The sequence of Figure 2 and Figure 3 has been changed.
Point 4: Summation with index i in formaula 5 line 159 is not clear as there is no index i inside.
Response 4: The index i in formaula 5 has been deleted in line 160 of the revised manuscript.
Point 5: line 218: "reference-free" but figure: "residual-free".
Response 5: Spelling mistake has been fixed in line 221 of the revised manuscript.
Point 6: m is not defined in line 200.
Response 6: m is defined in line 203 of the revised manuscript.
Point 7: line 214: "impressive performance gains" are to be calculated and given in numbers.
Response 7: Compared with residual-free model, our dilated multilevel network significantly improves the fusion performance, the performance-to-epoch curves (ERGAS) displayed in Figure 1 show the improvement of our model.
Point 8: line 222: WorldView-2 but WorldVorld-2 in line 168.
Response 8: Spelling mistake has been fixed in line 175 of the revised manuscript.
Point 9: line 239 defines L with tilde as MSE but there is also L without tilde in 7 that is MSE.
Response 9: L without tilde is MSE term, and the error has been changed in line 243 of the revised manuscript.
Point 10: "max" in formula 10 line 248 should not be italized.
Response 10: The “max” has been modified in line 252 of the revised manuscript.
Figure 1. Comparison results of the proposed model and residual-free network.

Reviewer 3 Report
This paper really makes a great contribution in related research field, but there is still need to be improved in the “Conclusion”. The “Conclusion” is too weak; the authors should try to reinforce it. For example, the contributions to academic research as well as theoretical implications and research limitations.
Author Response
Response to Reviewer 3 Comments
Dear reviewer,
Please find our revision to Algorithm, “learning an efficient convolution neural network for pansharpening”, with manuscript number: algorithms-412537.
We have made revision to the paper taking into account the reviewer’s comments. Please find our point-by-point response to the comments from the next page.
We thank the reviewer for helpful comments about our approach. We have tried to address everything satisfactorily in this revision.
Thanks,
Fei Ye
Corresponding Email: feiye.chn@gmail.com
This paper really makes a great contribution in related research field, but there is still need to be improved in the “Conclusion”. The “Conclusion” is too weak; the authors should try to reinforce it. For example, the contributions to academic research as well as theoretical implications and research limitations.
Response: We have reinforced our “Conclusion” by adding the research limitation of our work, and future work in line 375 of the revised manuscript.
Conclusion
In this paper, we propose an efficient model motivated by three goals of pansharpening: spectral preservation, spatial enhancement and model robustness. For the spectral and spatial domains, we employ an end-to-end CNN model that breaks the limitation of linear pansharpening algorithms. Experimental results demonstrate that our model tends to return well-balanced performance in spatial and spectral preservation. For the improvement of robustness, our CNN model is shallow while efficient, which can be less prone to overfitting. By adopting dilated convolution, our model achieves larger receptive field, and greatly makes up the shortcoming of network depth. Among the model, we also employ a multilevel structure to make full use of the features extracted under different receptive fields. Compared with state-of-the-art algorithms, the proposed model makes a better trade-off between spectral and spatial quality as well as generalization across different satellites.
Our model is motivated to be robust when generalized to new satellites, which is designed under the guiding concepts of simplicity and efficiency. As the generalization is improved, the shallow network architecture also restricts the fusion accuracy. In our future work, we will stick to the fusion of one specific satellite dataset, in that case, we can remove the effect of generalization, and focus on optimizing the architecture of a deeper network to further boost the fusion performance.
Please refer to the "Response.pdf" for detailed review response.

Round 2
Reviewer 1 Report
The authors have answered all queries to satisfaction.
Reviewer 3 Report
This revision has met all of my requirements in quality.